# Longitudinal Insights from Blended Hearing Care: Service Modality Choices, Support Received, and Satisfaction Ratings

**DOI:** 10.3390/healthcare13070689

**Published:** 2025-03-21

**Authors:** Sophie Brice, Justin Zakis, Helen Almond, Stefan Launer, Charlotte Vercammen

**Affiliations:** 1Institute of Health and Management, 185-187 Boundary Road, North Melbourne, VIC 3051, Australia; helen.almond@ihm.edu.au; 2Australian Institute of Health and Service Management, COBE, University of Tasmania, Hobart, TAS 7001, Australia; 3Sonova Audiological Care Australia, Melbourne Office, Carlton, VIC 3053, Australia; 4Sonova AG, Audiology & Health Innovation, 8712 Stäfa, Switzerland; stefan.launer@sonova.com (S.L.); charlotte.vercammen@sonova.com (C.V.); 5School of Health and Rehabilitation Sciences, The University of Queensland, Brisbane, QLD 4072, Australia; 6Manchester Centre for Audiology and Deafness, School of Health Sciences, Faculty of Biology, Medicine and Health, University of Manchester, Manchester M13 9PL, UK; 7Department of Neurosciences, Research Group Experimental Oto-Rhino-Laryngology, KU Leuven—University of Leuven, 3000 Leuven, Belgium

**Keywords:** chronic health conditions, hearing loss, behavioural interventions, telehealth, blended service delivery, service modality choices, satisfaction ratings

## Abstract

**Background/Objectives:** Sensorineural hearing loss (HL) is a highly prevalent chronic health condition. It can be managed through hearing care, including the use of hearing aids (HAs). Still, a majority of individuals with HL remain undiagnosed or untreated. Virtual care delivery may support uptake and adherence to interventions. In blended care, individuals can choose interchangeably between in-person and virtual services. This study aimed to investigate how real-world individuals accessed blended hearing care (through in-person, virtual, or hybrid services), the amount of support they received, and their satisfaction with services and products. **Methods**: An exploratory, retrospective analysis was performed on longitudinal observational data collected through Australia’s longest-running blended hearing care model. A total of 25,058 appointment records were available, matched to HA purchase records and clinical notes where possible, as well as 916 satisfaction ratings. **Results**: The majority of individuals attended in-person appointments (75%); 25% were virtual or hybrid appointments. The number of appointments attended depended on how HAs were purchased (in-person, virtually, or hybrid), but all modalities were complemented by ample unscheduled email and telephone support. Of those who purchased HAs repeatedly, 49% changed preferred sales channel (in-person versus virtual) over time. Satisfaction ratings were highest for virtual services. **Conclusions**: This first report of real-world, longitudinal evidence on blended hearing care showed strong attendance of in-person appointments, while hybrid services—including informal; unscheduled support—may have responded to individuals’ changing needs and preferences over time. The findings offer practice-based evidence for blended care models and recommendations for further research.

## 1. Introduction

Hearing care aims to increase the quality of life and well-being of individuals with hearing loss (HL) [1,2]. One component of hearing care is the provision of hearing aids (HAs)—medical devices designed to amplify and enhance incoming sounds; thereby increasing access to auditory information and supporting effective communication [3,4]. Hearing loss (HL) management, including the adoption and use of HAs, is a behavioural health intervention. It is recommended that HAs are worn daily, ideally during all waking hours, to achieve and maintain optimal benefit [5,6,7]. As such, successful use is directly dependent on the person’s choice to adopt and wear HAs consistently. HA uptake is generally considered to be low, with more than 80% of individuals with HL remaining undiagnosed or untreated [8,9].

Virtual service delivery models have been proposed to enhance the uptake and adherence to hearing care by addressing access barriers [10,11,12,13], saving travel time and costs [14,15]. Virtual care is delivered remotely using information technologies, such as videoconferencing tools, smartphones, and telephones [16,17]. Evidence supports the effectiveness and safety of virtual services in audiology [18,19]—often referred to as teleaudiology; eAudiology; remote; or connected hearing care [20]. Survey studies report that 30–40% of individuals accessing hearing care express an interest in virtual services [21,22]. Despite this interest and the evidence on virtual services, they have not been consistently offered, nor did their utilisation increase to the expected degree during the COVID-19 pandemic, when clinic-based care experienced operational challenges [23,24,25]. Kelsall-Foremann et al. [25], for example, reported only 12% of Australian survey respondents had been offered virtual hearing care services at the peak of the pandemic [25]. Of those who were offered and had accepted virtual services, though, 79% reported they were likely or very likely to recommend them [25]. These findings suggest it might be those providing rather than receiving hearing care services that act as a barrier to the uptake of virtual care—a challenge observed in other health domains as well [26,27,28].

It is imperative to differentiate between service providers who view virtual services as a supplementary offering and those who seamlessly integrate them into their operational framework. Blended care models are an example of the latter [29,30]. In blended care, clinicians are equally trained and equipped to deliver in-person and virtual services. The actual service modality choice is informed by clinical practice guidelines and the needs and preferences of the individual receiving care [29,30]. To date, however, longitudinal evidence on real-world use of hybrid hearing care services is scarce. It is unclear how services are most commonly provided in these models—in-person; virtually; or hybrid—if service modality choices change over time; and how individuals receiving care experience them. A better understanding of real-world experiences with blended care models may guide recommendations and further research aimed at increasing uptake of and satisfaction with hybrid services, ultimately supporting intervention adoption overall. Insights into satisfaction are of particular interest in healthcare fields such as audiology, where individuals seek healthcare and access services, and are also consumers of hearing-related products, such as HAs.

Between 2008 and 2021, a fully blended hearing care service delivery model was run in Australia. For over a decade, individuals could access hearing care services, including HA provision and support, through in-person (in-clinic), virtual (remote), or hybrid services (a combination of in-person and virtual services). Appointments were scheduled in advance. Individuals with more complex needs, such as those requiring custom-made earpieces for HAs to optimally fit, were triaged for in-person appointments in accordance with clinical practice guidelines [17,30,31,32]. Technologically proficient individuals were triaged for virtual services when this matched their personal preferences [30]. In addition to scheduled appointments, all individuals—independent of whether they attended appointments in-person or virtually—could access information; resources; and personalised support via additional; unscheduled communications. The latter were provided virtually—via email, telephone, or mobile application (video call or messenger services). A typical HA consumer journey consists of the following stages, all of which could take place in-person or virtually:Informational Guidance: Individuals received advice based on clinical hearing tests (for a detailed overview of the test, triage, and referral procedures, see [29,30]).HA Purchase and Trial Period: Guided by clinical recommendations, individuals could select HAs, purchase them, and begin an HA trial period.HA Fitting: HAs were fitted in-person at the time of purchase or shipped to the individual’s home for virtual HA fitting.Review and Adjustment: HA reviews or adjustments were conducted within the first few weeks, either virtually or in-person, regardless of the initial fitting method.Troubleshooting: Guided repair screenings and maintenance assistance were offered as requested or required.Trial Decision: Individuals decided to keep or return the HAs within the typical 30-day trial period.

During the thirteen years of clinical operation of this blended hearing care model, the longest running in Australia to date, data were collected routinely as part of ongoing service delivery. This study describes an exploratory, retrospective analysis of the observational data collected (see Methods section for full details). Thereby, we aimed to investigate how real-world individuals accessed blended hearing care (through in-person, virtual, or hybrid services), the amount of support they received (appointments attended, as well as informal support received via email or telephone), and how individuals rated their satisfaction with blended hearing care. We considered the following research questions (RQs):

Service modality choices:
RQ1a: What service modalities were chosen for scheduled appointments when individuals could opt for both in-person and virtual services for any part of their hearing care?RQ1b: What sales channels were chosen when individuals could purchase HAs in-person or fully virtually through the website, thereby starting an HA trial period? For those who adopted and purchased HAs multiple times, did they consistently prefer the same channel?


Support received:
RQ2a: Did the number of appointments scheduled by individuals seeking hearing care depend on the chosen sales channel, i.e., on whether they had purchased HAs in-person or virtually?RQ2b: Did the amount of additional, unscheduled support provided to individuals seeking hearing care depend on the chosen sales channel, i.e., on whether they had purchased HAs in-person versus virtually?

Satisfaction ratings:
RQ3: Did satisfaction with HAs and hearing care services depend on individuals’ chosen sales channel, i.e., on whether they had purchased HAs in-person or virtually?

## 2. Materials and Methods

### 2.1. Data Sources

Between 2008 and 2021, real-world, observational data were collected routinely as part of ongoing service delivery of Australia’s longest-running blended hearing care model [33]. The data were collected through (1) Customer Management Software (https://www.worketc.com/), in which appointment records and clinical notes related to individual clients were stored; (2) a sales database, in which HA purchase records of individual clients were logged; and (3) satisfaction surveys, solicited by an independent third-party provider [34] who sent emails to individual clients within three weeks of HA purchase to ask about their experiences with the services received and products purchased.

We were granted access to the data by the companies behind the blended hearing care model (Blamey Saunders hears, later Sonova AG). To this end, the data were extracted from the three data sources by the blended hearing care model’s Information Technology Department in the form of Microsoft Excel data export tables. All personal identifiers were removed from the data prior to export. Data were considered for inclusion in the data export tables based on their availability: Migration to different database systems for the Customer Management Software and HA sales database resulted in data loss, which meant that data were only available for certain periods of time depending on the data source (see Figure 1 for an overview). Data belonging to non-Australian residents were excluded (as individuals based outside of Australia did not have access to in-person services), as well as sales records that belonged to products other than HAs (such as HA accessories). The final selection of de-identified eligible data were then securely stored on a protected server with access granted only to the researchers. The exported data tables were imported into R statistical software version 4.4.1 [35] and Minitab Version 17 for further data cleaning and data analysis.

### 2.2. Data Cleaning

As the data were not collected for research purposes but exported from clinical and business databases; extensive data cleaning was required to prepare the data for further analysis.

Customer Management Software. The Customer Management Software contained two types of data: (1) appointment records and (2) clinical notes (see Figure 1).

Appointment records available to us were logged between May 2011 and August 2019 (see Figure 1) and contained unique consumer codes, time stamps, scheduled appointment durations, and appointment codes. The appointment codes were assigned at the time of booking. They were updated by a clinician at the time of the appointment to reflect whether the appointment was attended or not and in what modality services were provided (in-person or virtually). Individuals with incomplete appointment histories were excluded from further analyses. In some instances, modality information, i.e., whether an appointment was attended in-person or virtually, was missing from the raw data, while the appointment history was otherwise complete. These data points were considered for further analysis.

Clinical notes available to us were logged by clinicians in the Customer Management Software between May 2011 and August 2019 (see Figure 1). As part of their note taking, clinicians kept track of unscheduled support provided to individuals, i.e., support provided beyond pre-booked appointments. Unscheduled support records included a time stamped history of emails (i.e., single emails that could be inbound or outbound) and notes (i.e., single notes created by a clinician in response to contact, such as a telephone or video call conversation, to summarize a support session). Due to the structure of the database, extracting unscheduled support types for each individual could not be automated. Therefore, a random sampling of complete contact history for forty individuals was manually extracted from the database. Half of these forty individuals had purchased HAs in-person, while the other half had purchased HAs virtually. Unscheduled support records could be matched to booked appointment information through the shared use of unique consumer codes.

Sales database. HA purchase records available to us were logged between March 2016 and September 2019 (see Figure 1) and contained unique consumer codes, time stamps, and sales channel information, i.e., whether an HA was purchased in-person or virtually. The sales channel was recorded by a clinician at the time of purchase. Incomplete purchase records were excluded from further analyses. HA purchase records could be matched to booked appointment information through shared use of the unique consumer codes.

Satisfaction surveys. Between 2017 and 2021, satisfaction survey data were collected in the form of ratings (see Figure 1). These ratings were solicited by an independent, third-party provider [34] who sent emails to consumers during the trial period, i.e., within three weeks of HA purchase. Individuals were asked to rate how likely they would recommend the clinical services received and/or HAs purchased to family or friends on a discrete scale from one to five. The satisfaction data also contained the sales channel (i.e., virtual versus in-person), date of purchase, date of receipt of consumer review, and HA type/model. Only complete data entries were considered. Since the records did not include the unique consumer codes used in other databases, the data could not be linked to the appointment booking software.

### 2.3. Data Analysis

Data were analysed descriptively for all RQs. For RQ2a and RQ2b, hypothesis testing was employed. Based on visual inspection of the data and assumption testing, non-parametric alternatives were employed where appropriate.

In RQ3, we explored satisfaction with HAs and hearing care services through ratings given on a scale from one to five. These ratings were used to calculate a Net Promotor Score^TM^ (NPS), a metric first introduced by Reichheld [36,37] as a means for businesses to gauge consumer loyalty [36,37]. Individuals who provided ratings with a score of one to three were categorised as “Detractors”, four as “Passives”, and five as “Promotors”. NPS outcomes were then derived by deducting the percentage of detractors from the percentage of promotors [36,37]. Confidence intervals for the NPS outcomes were calculated based on the adjusted Wald method, variation AW (3, T), as described by Rocks in 2016 [38].

## 3. Results

### 3.1. Service Modality Choices

In RQ1a, we explored appointment modalities, i.e., whether individuals attended scheduled clinical appointments in-person or virtually. Data extracted from appointment booking software revealed 25,058 appointment records, belonging to 6864 individuals. Table 1 provides an overview of the number of appointments scheduled per appointment modality; 82% (n = 20,633) of appointments took place as in-person visits and 8% (n = 2051) as virtual appointments. Appointment modality information was not available for 9% (n = 2374) of appointments (Table 1, rows 3–5). Close to half of the appointments with missing modality information (n = 1112) were marked as cancelled appointments (Table 1, row 3); in a minority of cases, the consumer had not shown up for the appointment (n = 126; Table 1, row 4).

After removing the cancelled and no-show appointments (4.9%; n = 1238), the data set contained data of 6766 unique individuals. They attended a median number of 2 appointments (IQR: 1–4; min: 1; max: 36). Table 2 provides an overview of appointment modalities. Results showed that 74% (n = 5024) of individuals attended exclusively in-person appointments, 22% (n = 1477) hybrid appointments, i.e., a mix of in-person and virtual appointments, 3% (n = 176) exclusively virtual appointments, and 1% (n = 89) had no appointment modality information available for their appointments. For the latter classification, appointments with missing appointment modality information were removed. For example, if someone attended six appointments, five of which were in-person and one appointment without known modality information, the individual was classified as attending exclusively in-person appointments. Individuals whose appointment modality information was missing completely were classified under “no information available”. After removing appointment records without appointment modality information, the data set contained data of 6677 unique individuals, and the percentages were similar: 75% (n = 5024) attended exclusively in-person appointments, 22% (n = 1477) hybrid appointments, and 3% (n = 176) exclusively virtual appointments.

In RQ1b, we explored chosen sales channels, i.e., whether individuals chose to purchase HAs in-person or virtually through the website. Data extracted from the HA purchase records revealed 1523 complete sales records belonging to 1361 individuals. Based on available sales records, 60% (n = 919) of HA purchases were finalised in-person, while a further 40% (n = 604) were finalised virtually through the website. Of 1361 individuals, 58% (n = 786) preferred to purchase HAs exclusively in-person, and 37% (n = 506) preferred purchasing virtually. A minority (5%; n = 69) had purchased HAs through a mixture of sales channels (“hybrid”), i.e., they purchased multiple times, sometimes in-person and sometimes virtually, with the first sale taking place in-person for most (n = 51) and virtually for some (n = 18).

Figure 2 visualises chosen sales channels for all 142 individuals who had gone through multiple purchases: 142 individuals purchased HAs on two separate occasions, 18 individuals on three occasions, and two individuals on four occasions. Overall, 32% (n = 46) purchased HAs exclusively in-person, 19% (n = 27) exclusively virtually, and 49% (n = 69) in a hybrid manner. In other words, 49% of return buyers chose different sales channels over time, while 51% exclusively chose one sales channel. For a first sale, 68% (n = 97) of purchases were finalised in-person as opposed to 32% (n = 45) virtually. For a second sale, 49% (n = 69) of purchases were finalised in-person as opposed to 51% (n = 73) virtually.

### 3.2. Support Received

In RQ2a, we explored the number of appointments scheduled and attended by individuals. In particular, we investigated whether the number of appointments depended on the chosen sales channel, i.e., on whether individuals purchased HAs exclusively in-person, exclusively virtually, hybrid with the first sale taking place in-person, or hybrid with the first sale taking place virtually. After excluding cancelled and no-show appointments, data from the appointment booking software were matched to sales records. This revealed data of 1215 individuals, who attended 3701 appointments, and for whom 1370 sales were recorded. This dataset was slightly smaller than the one considered for RQ1b, as some unique consumer codes were not available in both the appointment and sales records. Figure 3 provides an overview of the number of appointments booked per individual as a function of their chosen sales channel. When someone purchased HAs on more than one occasion, they likely scheduled more appointments than someone who purchased HAs once. Therefore, we normalised the number of appointments for each individual by dividing them by their respective number of HA purchases. For simplicity, the normalised number of appointments is further referred to as appointments. Across the 1215 individuals, 755 (62%) purchased HAs exclusively in-person (they did not purchase virtually). They attended a median of 3.0 appointments (IQR: 1.0–4.0, min: 0.5, max: 15.0). A further 393 (32%) individuals purchased exclusively virtually (they did not purchase in-person). They attended a median of 2.0 appointments (IQR: 1.0–3.0, min: 0.5, max: 10.0). Another 49 (4%) individuals had purchased in a hybrid manner, i.e., both in-person and virtually, with the first sale taking place in-person. They attended a median of 1.5 appointments (IQR: 1.0–2.0, min: 0.5, max: 6.0). Finally, 18 (1%) individuals had purchased in a hybrid manner, with the first sale taking place virtually. They attended a median of 2.3 appointments (IQR: 1.5–3.4, min: 0.5, max: 4.5).

Shapiro–Wilk testing (*W* (1214) = 0.81, *p* < 0.0001) suggested that the distribution of appointments was significantly different from a normal distribution. This was in line with the visual inspection of the data. A non-parametric Kruskal–Wallis test showed that the appointments significantly depended on the chosen sales channel (*H* (3) = 51.70, *p* < 0.0001). Post-hoc Dunn tests with Bonferroni-corrected *p*-values showed that individuals who purchased HAs exclusively in-person attended significantly more appointments than those who purchased HAs exclusively virtually (*z* = 5.48, *p* < 0.0001, *r* = 0.16) and those who used hybrid sales channels, with their first purchase taking place in-person (*z* = 5.35, *p* < 0.0001, *r* = 0.15). Individuals who purchased HAs virtually also attended significantly more appointments than those who used hybrid sales channels with their first purchase taking place in-person (*z* = 2.95, *p* < 0.05, *r* = 0.08). There were no significant differences in appointments between individuals who used hybrid sales channels with their first purchase taking place virtually and other chosen sales channels, i.e., those who purchased HAs exclusively in-person (*z* = −1.28, *p* = 1.00, *r* = −0.04), exclusively virtually (*z* = 0.15, *p* = 1.00, *r* = 0.004), or who used hybrid sales channels with their first sale taking place in-person (*z* = 1.75, *p* = 0.48, *r* = 0.05).

In RQ2b, we explored the amount of additional, unscheduled support provided to individuals via email, telephone, or mobile application. In particular, we investigated the number of Emails sent (i.e., inbound or outbound) and Notes created by clinicians (i.e., in response to telephone or video call conversations). We also investigated whether these types of unscheduled support depended on the individuals’ chosen sales channel, i.e., on whether HAs were purchased in-person or virtually (see Figure 4). As described in detail in the Data Extraction section, the full contact history including unscheduled support data were manually sampled from multiple databases for forty individuals, as the process could not be automated. Half of the individuals had purchased their HAs in-person, while the other half had purchased HAs virtually.

When an individual purchased HAs on more than one occasion; they likely requested more unscheduled support than an individual who purchased HAs once. Therefore, we normalised the number of unscheduled support contacts for each individual by dividing them by their respective number of HA purchases. For simplicity, the normalised number of unscheduled support contacts is further referred to as unscheduled support contacts. Individuals experienced a median of 24 unscheduled support contacts per person (IQR: 16.8–30.1; min: 6.0; max: 49.0), i.e., a median of 10.8 Emails (IQR: 6.9–16.1; min: 0.5; max: 34.5) and 12.8 Notes (IQR: 8.5–15.4; min: 4.0; max: 31.0). Shapiro–Wilk tests suggested that the normalised Email data were normally distributed (*W* (*39*) = 0.95, *p* = 0.09), while the normalised Notes data were significantly different from a normal distribution (*W* (*39*) = 0.91, *p* < 0.01). Therefore, a balanced ANOVA was performed for Emails, while a Kruskal–Wallis test was performed for Notes, with sales channel (In-person versus Virtual) as independent variable. The number of Emails (*F* [1,38] = 3.95, *p* = 0.05) and Notes (*H* (1) = 0.15, *p* = 0.69, *r* = −0.06) were not significantly different between consumers who purchased in-person or virtually.

### 3.3. Satisfaction Ratings

In RQ3, we explored satisfaction with HAs and hearing care services through ratings, given on a scale from one to five. A total of 916 service ratings and 905 product ratings were available, both collected following HA purchases. Of those, 643 (70%) purchases took place in-person, and 273 (30%) purchases took place virtually. Those who purchased HAs in person gave median satisfaction ratings of 5.0 for products (IQR: 4.0–5.0, min: 1.0, max: 5.0) and 5.0 for services (IQR: 5.0–5.0, min: 1.0, max: 5.0). Those who purchased HAs virtually gave median satisfaction ratings of 5.0 for products (IQR: 4.8–5.0, min: 1.0, max: 5.0) and 5.0 for services (IQR: 5.0–5.0, min: 1.0, max: 5.0). While the descriptive results suggest a ceiling effect, the raw satisfaction data were also used to calculate NPS outcomes (see Methods section). Table 3 provides an overview of NPS outcomes together with 95% confidence intervals (95%CI) for services and products separately and as a function of sales channels (in-person versus virtually).

## 4. Discussion

In this study, we explored real-world longitudinal data, collected routinely as part of Australia’s longest-running blended hearing care model. To the best of our knowledge, it is the first report to provide practice-based evidence on (1) how individuals sought hearing care and purchased HAs when offered hybrid services, (2) the amount of scheduled and unscheduled support individuals received, and (3) how satisfied individuals were with the corresponding services and products. While the context of this report is hearing care in Australia, the findings could benefit virtual or hybrid hearing health services globally by providing recommendations to guide future research into more accessible and effective hearing rehabilitation services.

### 4.1. Service Modality Choices

Three-quarters of the 6677 (75%; n = 5024) individuals in this sample attended in-person appointments. A further 25% (n = 1653) attended virtual or hybrid services (RQ1a). These numbers were lower than the 30–40% of audiology consumers who indicated they were interested in virtual services in survey studies [21,22]. What is noteworthy is the blended service delivery model provided additional support to all individuals—regardless of scheduled appointment modalities—through a dedicated teleaudiology team who responded to all unscheduled communications via email; telephone; or mobile application (also see RQ2b). The availability of the teleaudiology support team implies that even when scheduled appointments were highly attended in-person, individuals could still rely on very accessible virtual services. Also, not all individuals who might have preferred virtual appointments would have been eligible for them. Individuals with complex health or audiological needs would have been triaged towards in-person services, guided by clinical practice guidelines [31,32]. Results further revealed that a greater proportion of the individuals received virtual or hybrid services (25%; n = 1653 individuals) than the overall proportion of virtual services indicated (8% of all appointments were attended virtually; n = 2051). These results suggest data on service modality choices may be more informative from a longitudinal individual perspective than from a service-orientated view. Health management is inherently a longitudinal process, during which individual needs and preferences can evolve over time [39,40]. Prior research, however, has tended to consider preferences for virtual services cross-sectionally, i.e., by asking how many individuals were interested in virtual care at a certain point in time or reporting on the proportion of individuals who had any experience with such services in the past [22,24,25].

Sales records showed that HAs were purchased in-person in the majority of cases (60%; n = 919), as opposed to through the website (40%; n = 604). Longitudinal trends—for 142 individuals who adopted HAs multiple times—showed that the proportion of virtual sales increased from 32% (n = 45) for a first purchase to 51% (n = 73) for a second purchase. These findings suggest that individual needs and preferences can and do evolve since close to half of these individuals (49%; n = 69) demonstrated a change in preferred sales channel over time (see Figure 2; RQ1b). To the best of our knowledge, this is the first report of longitudinal service modality choices for hybrid hearing care services. Most service providers offer only one service modality: exclusively in-person or exclusively virtual services [41], or HA trial periods are always started in-person, even when virtual follow-up care is available [42]. Blended service delivery models are uniquely positioned to investigate individual choices and preferences, as clinicians are equally trained and equipped to deliver in-person and virtual services for any aspect of blended hearing care [30]. Findings also suggest that factors driving preferences might be more complex than, e.g., mobility and transportation concerns—often reported as key consumer motivations to consider virtual care [43]. An earlier report on blended models indicated that individual distance to a clinic had variable relationships to purchase modality preferences [44]. Individuals living in close proximity to a clinic still pursued virtual services, and individuals travelling far beyond 70 km still chose to travel to a clinic [44]. We did not report on distance or proximity to a clinic here, although the observation of individuals changing purchase modality over time, sometimes even back and forth (see Figure 2), indicated that location alone cannot explain such preferences.

### 4.2. Support Received

In this sample, individuals who exclusively purchased HAs in-person attended significantly more appointments than individuals who purchased HAs exclusively through the website, even after correction for the number of sales (RQ2a, see Figure 3). We could hypothesise that this is due to the complexity of individual needs. Individuals with the greatest need complexity may have required more support and, consequently, more appointments. Based on clinical practice guidelines, they would have been triaged toward in-person services [17,30,31,32]. Individuals who adopted HAs multiple times, in-person first and later at least once virtually, attended significantly fewer appointments than individuals who purchased exclusively in-person or exclusively through the website (also see Figure 3). Certain individuals, such as experienced HA wearers, might have been triaged towards virtual services by their clinician, especially for a second or third HA purchase and when it matched their personal preferences. We could hypothesise these customers may have developed self-efficacy over time. Self-efficacy can broadly be described as the extent to which a person has confidence in their own abilities to make a plan and execute it in order to achieve a certain goal [45]. Within hearing care, these goals can be related to understanding speech in various listening situations (“listening self-efficacy”) [46,47] or to managing, handling, and adjusting the HA [47].

In addition to scheduled appointments and support provided, unscheduled support information was explored for a subset of forty individuals: We looked into how many emails were sent (inbound or outbound) and how many clinical notes were generated following an unscheduled, virtual support session. While there was a trend towards group differences for emails, results suggested clinicians recorded an equal number of unscheduled support for all individuals, independent of whether they purchased HAs in-person or virtually (RQ2b). It is possible that too few data points were available to pick up differences between groups. However, the overall amount of unscheduled support provided to individuals—independent of sales channel—might be the most revealing outcome (RQ2b). While individuals attended a limited number of scheduled appointments, between 1.5 and 3 appointments depending on sales channel (RQ2a), clinicians provided ample unscheduled support to individuals, as evidenced by 11 emails and 13 notes documented per individual. To the best of our knowledge, this is the first report of unscheduled support provided to individuals seeking hybrid hearing care services. Many clinics may provide similar informal support via various administrative roles, with varying degrees of integration within clinical scheduling. A small business with a sole practitioner managing all clinical and administrative communications may find scheduled appointments under-represent the amount and degree of support provided for in-person services as well.

In this model, clinician interactions were highly accessible. All individuals, independent of whether they attended appointments in-person or virtually, could easily receive additional support without planning for it. A dedicated support team received all unscheduled communication via email, telephone, smartphone-based video calls, or messenger services. Innovations in virtual health technologies have indeed impacted hearing care by providing a multitude of new and emerging possibilities and practices [48,49]. It is, for instance, common for HA users to experience at least one issue following an HA fitting [50], and follow-up support can be crucial for mitigating the risk of cessation of HA use [51,52]. Resolution of issues via virtual services has been shown to be comparable to in-person services [53]. For in-person services, lack of disclosure of issues by individuals was found to be an important factor, ultimately risking continued HA use [50,52,54]. The large number of unscheduled virtual support types identified here is encouraging and may imply that virtual health technologies may lower the threshold to disclose support needs. At the same time, findings raise the need to consider the training and expertise of personnel chosen to respond to those support needs.

### 4.3. Satisfaction Ratings

Five-item satisfaction ratings were converted to NPS outcomes (Table 3). While the use of NPS, a single-item metric, has been criticised, particularly in terms of its validity in healthcare settings [55], it is becoming increasingly common to measure and report on it, including in hearing care [41,56,57]. Satisfaction and experiences of those seeking healthcare are often considered key healthcare performance metrics, along with patient safety and clinical effectiveness measures [58]. Yet, there is no consensus on how to measure satisfaction in a healthcare context [59]. As NPS data can be collected in a quick and simple manner [53], they have been used as a proxy for satisfaction [41].

Overall NPS outcomes provide an indication of how a service provider is performing as a whole [60]. The overall result reported here, i.e., an NPS of 69 [95%CI: 66–71], was higher than NPS outcomes for other healthcare providers in Australia, such as for general practices, dentists, and pharmacists [61,62]. For hearing care, the current NPS outcomes were comparable to the NPS reported for a virtual hearing care delivery model in the United States (NPS of 66) [41], yet lower than the NPS of a South African-based model offering hybrid services (NPS of 87) [56], keeping in mind that the latter study only had 31 data points. At subgroup levels, NPS outcomes were higher in this sample for individuals who purchased HAs virtually (NPS of 79 [95%CI: 74–83]) compared with in-person (NPS of 65 [95%CI: 61–68]). The high NPS outcomes for virtual sales may have been due to several factors, such as triaging of consumers with more complex needs towards in-person care and triaging of more technically proficient individuals to virtual services. NPS outcomes based on the 2022 MarkeTrak surveys showed the opposite patterns, with higher NPS scores for in-person (NPS of 30) compared with virtual HA fittings (NPS of -4) [60]. Variable applications of the term virtual services may be of key importance in interpreting such differences, requiring clarity on how services were actually provided and within what service delivery model [10]. NPS outcomes in this sample were higher for clinical services (NPS of 78 [95%CI: 75–81]) than for products (NPS of 59 [95%CI: 55–63]).

At an individual level, simple measures of satisfaction can be leveraged to trigger pro-active support to those who may need it most [34], as was implemented in this service delivery model. Satisfaction has long been identified as a potential indicator of the likelihood of adhering to chronic disease management [63,64,65,66,67]. Also in hearing care, satisfaction metrics are considered useful clinical measures of patient-reported outcomes [41,56,68], further supported by their correlation with continued HA use [69] and daily HA wearing time [69]. Future research on potentially contributing factors would be required to provide insights that could impact clinical service design. Fuentes-Lopez [70], for instance, reported a significant relationship between self-efficacy and average daily hours of HA use [70]. Given the data reported here, it would be of interest to investigate if individuals who gave higher satisfaction ratings also had higher self-efficacy and were more likely to wear their HAs more consistently.

## 5. Limitations and Recommendations

It is important to highlight that the data presented here were not collected for research purposes. The data were collected routinely as part of ongoing hearing care service delivery and analysed retrospectively, which may have impacted the quality of the data and generalisability of the findings [71,72]. Migration to new database systems over the years, for instance, led to a loss of data. Also, the dataset contained little demographic data, a limited number of therapeutic outcome measures only, and no information on how often HAs were returned after purchase—all of which would be important to better understand outcomes with hybrid hearing care services. However, to the best of our knowledge, this is the first report of a large, real-world, longitudinal dataset detailing how individuals accessed and experienced blended hearing care. The results offer practice-based evidence and may guide further research aimed at increasing uptake of and satisfaction with hybrid services and offering more accessible and effective hearing rehabilitation services. Real-world evidence can indeed foster continuous improvements in health care provision and personalised care [73,74,75]. Such improvements could be aimed at adapting existing service delivery models or designing new ones for contemporary health consumers and the improvement of health outcomes for all. Thereby, adaptations or designs would ensure those who need help can be supported effectively when they need it, while continuing to explore pathways to serve a larger array of individuals.

Based on the results presented, we propose future research to consider collecting longitudinal empirical data aimed at:Investigating how service usage and modality choices depend on clinical recommendations and individual preferences.Investigating changes in behaviours, preferences, and needs over time and throughout the consumer journey, including beyond the first few weeks of behavioural intervention.Identifying factors that might drive (changes in) behaviours, preferences, and needs.Considering objective and self-reported factors, as well as qualitative research methods, to better understand behaviours, preferences, and needs.Tracking informal, unscheduled support in addition to scheduled clinical support types.

## 6. Conclusions

In this study, the majority of individuals scheduled and attended exclusively in-person appointments (75%); 25% attended virtual or hybrid appointments. The number of support appointments scheduled depended on how individuals adopted HAs (in-person, virtually, or hybrid) and were moderate in number, whereas unscheduled support types (email or telephone-based) were omnipresent for all individuals. The latter finding is encouraging, as it may imply virtual health technologies lower the threshold to disclose support needs. Close to half the individuals who adopted HAs repeatedly changed purchase modality over time, sometimes even back and forth, suggesting individual needs and preferences changed over time. Satisfaction ratings were highest for virtual services.

This first report of real-world, longitudinal evidence on blended hearing care showed strong attendance of in-person appointments. Hybrid services—including informal; unscheduled support—may have responded to individuals’ changing needs and preferences over time. The findings presented offer practice-based evidence on blended care models and recommendations for further research, which may guide service delivery model design, clinical practices, and health policies. A clear recommendation to utilise and enable practice-based evidence in emerging clinical practices to inform research as it does clinical practice is of key importance.

## Figures and Tables

**Figure 1 healthcare-13-00689-f001:**
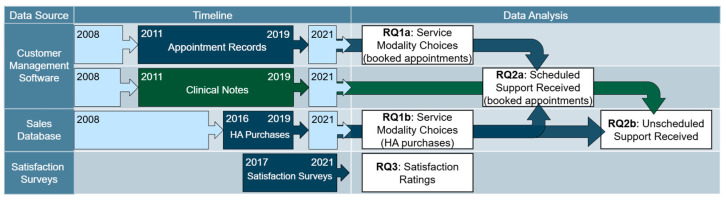
Schematic overview of data sources and timeframes during which the data were collected (left panel). Light blue boxes represent data that were unavailable for retrospective analysis due to changes in database management systems. Dark blue boxes represent data that were available for retrospective analysis. The dark green box represents data that were available but that could not be exported automatically; instead, manual sampling of the data were performed. The right panel provides an overview of what data were (combined and) used to address different research questions.

**Figure 2 healthcare-13-00689-f002:**
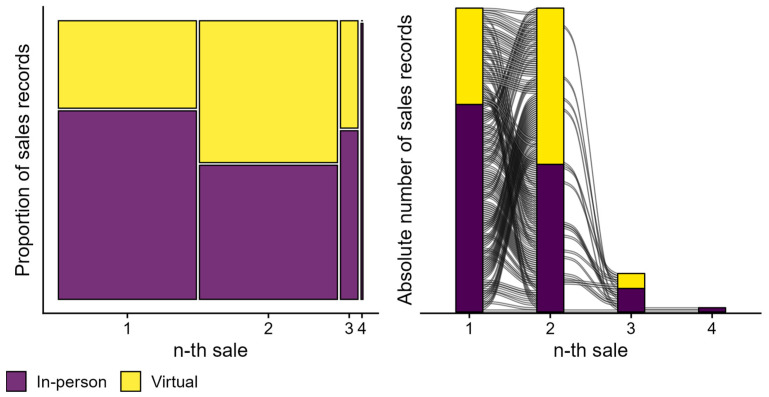
Overview of chosen sales channels for individuals who had purchased HAs more than once. The mosaic plot on the left visualises the proportion of sales (y-axis) finalised in-person (purple tiles) versus virtually (yellow tiles) for the n-th sale, i.e., the first, second, third, and fourth sales (x-axis). Column widths are proportional to the number of sales records available for every n-th sale, as plotted on the x-axis. The right panel visualises the absolute number of sales records (y-axis) as a function of sales number (x-axis). The alluvia represents individuals and how they moved between sales channels (purple and yellow bars).

**Figure 3 healthcare-13-00689-f003:**
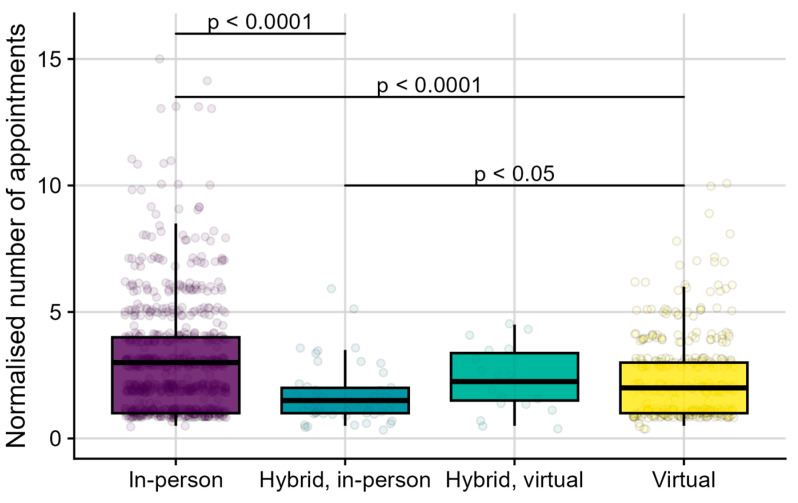
Boxplots of normalised number of appointments (y-axis) as a function of chosen sales channel (x-axis). From left to right, the chosen sales channels were “In-person”, for those who purchased HAs exclusively in-person; “Hybrid, in-person”, for those who used different sales channels over time, with a first sale taking place in-person; “Hybrid, virtual”, for those who used different sales channels over time, with a first sale taking place virtually; and “Virtual”, for those who purchased HAs exclusively virtually. Individual data points (bullets) show the normalised number of appointments for each individual, i.e., the number of appointments they attended divided by their respective number of HA purchases.

**Figure 4 healthcare-13-00689-f004:**
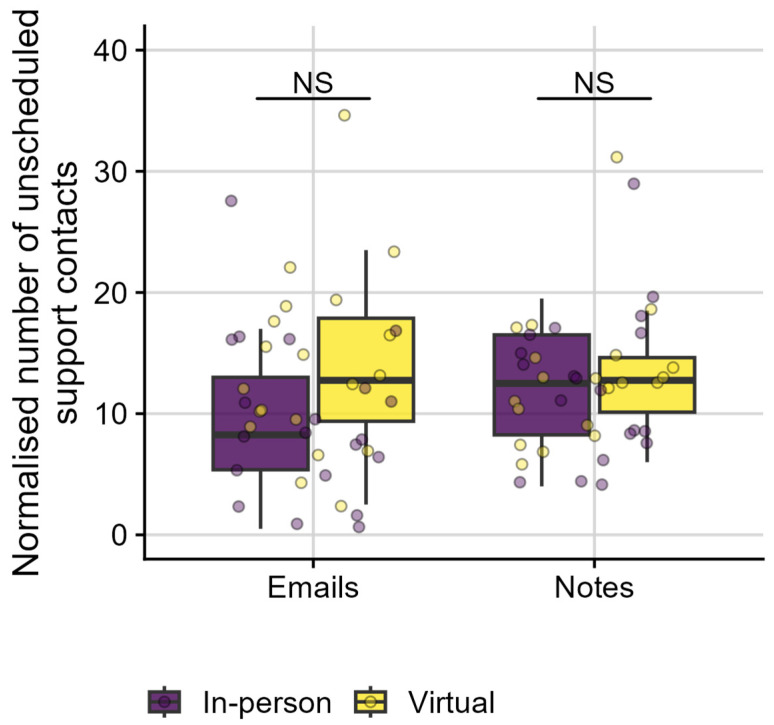
Boxplots of normalised number of unscheduled support contacts (y-axis) as a function of the type of unscheduled support provided, i.e., emails sent, inbound or outbound, and notes created by clinicians (x-axis). Individual data points (bullets) show the normalised number of unscheduled support contacts for each individual. Colour coding represents individuals’ chosen sales channels, with purple representing in-person purchases and yellow representing virtual purchases. NS indicates non-significant differences between groups.

**Table 1 healthcare-13-00689-t001:** Overview of number of appointments attended as a function of service modality for the total sample of 25,058 appointments extracted from the appointment booking software.

	Service Modality	Number of Appointments Scheduled	Percentage of Appointments Scheduled
1.	In-person	20,633	82.3%
2.	Virtual	2051	8.2%
3.	Cancelled	1112	4.4%
4.	No show	126	0.5%
5.	Missing	1136	4.5%

**Table 2 healthcare-13-00689-t002:** Overview of number of individuals as a function of service modality, for a total sample of 6766 individuals who had at least attended one appointment.

Appointment Modality	Number (Percent) of Individuals
Exclusively in-person	5024 (74%)
Exclusively virtual	176 (3%)
Hybrid	1477 (22%)
No information available	89 (1%)

**Table 3 healthcare-13-00689-t003:** Overview of NPS outcomes along with 95% CI in brackets, calculated based on satisfaction ratings for services and products separately, and as a function of sales channel, i.e., in-person versus virtually. Overall NPS outcomes were based on satisfaction ratings combined across sales channels (horizontally), across services and products (vertically), and across both (bottom right).

	In-Person	Virtual	Overall
Services	75 [70–79]	86 [81–90]	78 [75–81]
Products	54 [49–59]	71 [64–77]	59 [55–63]
Overall	65 [61–68]	79 [74–83]	69 [66–71]

## Data Availability

Data will be made available upon reasonable request due to restrictions.

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
