# Peer review of "Longitudinal Insights from Blended Hearing Care: Service Modality Choices, Support Received, and Satisfaction Ratings"

_healthcare, 2025, doi:10.3390/healthcare13070689_

Round 1

Reviewer 1 Report

Comments and Suggestions for Authors

The authors present the study "Longitudinal Insights from Blended Hearing Care: Service Modality Choices, Support Received, and Satisfaction Ratings"

The study topic about sensorineural hearing loss (SHL) is of important interest to science because even today, there are people without diagnosis and/or treatment.

The readers expect that the authors will close this manuscript, based on the results, proposing solutions to the dissatisfactions with combined hearing care and providing several suggestions and recommendations for future research.

Author Response

For research article

Response to Reviewer 1 Comments

1. Summary

Thank you very much for taking the time to review this manuscript. Please find our detailed responses below, as well as our corresponding revisions in the re-submitted files. We have consistently used a green font in the rebuttal letter and in the manuscript to mark our revisions and responses to the reviewer’s comments.

2. Questions for General Evaluation

Reviewer’s Evaluation

Response and Revisions

 See our reply to the point-by-point responses.

3. Point-by-point response to Comments and Suggestions for Authors

Comments 1: The authors present the study "Longitudinal Insights from Blended Hearing Care: Service Modality Choices, Support Received, and Satisfaction Ratings"

The study topic about sensorineural hearing loss (SHL) is of important interest to science because even today, there are people without diagnosis and/or treatment.

The readers expect that the authors will close this manuscript, based on the results, proposing solutions to the dissatisfactions with combined hearing care and providing several suggestions and recommendations for future research.

Response 1: Thank you for the valuable feedback. Investigation into factors associated with satisfaction outcomes was beyond the scope of this study, though the authors strongly agree such analyses would be of great value. In support of encouraging further research towards such questions, the recognition that further investigation and research is required and invited to explore factors beyond the scope of this study has been strengthened in the Limitations-, Recommendations- and Conclusions-sections:

·       Limitations and Recommendations: (p. 14, lines 552-569): “It is important to highlight that the data presented here were not collected for re-search purposes. The data were collected routinely as part of ongoing hearing care ser-vice delivery and analysed retrospectively, which may have impacted the quality of the data and generalizability of the findings [71, 72]. Migration to new database systems over the years, for instance, led to a loss of data. Also, the dataset contained little demographic data, a limited number of therapeutic outcome measures only, and no in-formation on how often HAs were returned after purchase—all of which would be important to better understand outcomes with hybrid hearing care services. However, to the best of our knowledge, this is the first report of a large, real-world, longitudinal dataset detailing how individuals accessed and experienced blended hearing care. The results offer practice-based evidence and may guide further research aimed at increasing uptake of and satisfaction with hybrid services, and offering more accessible and effective hearing rehabilitation services. Real-world evidence can indeed foster continuous improvements in health care provision and personalised care [73-75]. Such improvements could be aimed at adapting existing service delivery models, or designing new ones, for contemporary health consumers and the improvement of health outcomes for all. Thereby, adaptions or designs would ensure those who need help can be supported effectively, when they need it, while continuing to explore pathways to serve a larger array of individuals.”

·       Conclusions (p. 15, lines 597-601): “The findings presented offer practice-based evidence on blended care models and recommendations for further research, which may guide service delivery model design, clinical practices, and health policies. A clear recommendation to utilise and enable practice-based evidence in emerging clinical practices to inform research as it does clinical practice is of key importance.”

4. Response to Comments on the Quality of English Language

Point 1: The English is fine and does not require any improvement.

Reviewer 2 Report

Comments and Suggestions for Authors

Dear Authors,

I appreciate the opportunity to review your manuscript. The study addresses a relevant and highly interesting topic in the field of hearing care, especially in the context of hybrid care models. Below, I present a series of suggestions and recommendations that I believe could improve the clarity, depth, and relevance of the manuscript.

Abstract:
I suggest moving the study's objective to the Introduction section of the Abstract to make the study's purpose clear from the beginning. Additionally, include a brief mention of the practical implications of the findings in the conclusion of the Abstract.

Introduction:
I recommend expanding the justification of the study, explaining why it is important to investigate hybrid service modalities in the context of hearing care.
Please include an explanation of the practical implications of the findings. For example, how the results can influence the adoption of hybrid care models in clinical practice.
The objective is not sufficiently explicit. I recommend including it at the end of the Introduction.
I also suggest moving any information about data sources and methodology to the Methodology section. I believe the Introduction should focus on the context, relevance of the problem, and justification of the study. All information regarding data sources should be moved to the Methodology section.
I suggest highlighting more clearly that the study focuses on a hybrid hearing care model, mentioning both in-person and virtual services. This should also be reflected in the objectives.

Methodology:
I suggest detailing more clearly the specific inclusion and exclusion criteria for greater transparency.
More details should be provided about the statistical analysis procedures used, including any specific software.
I believe the ethical criteria of the research should be made explicit in this section. Was informed consent obtained from the participants, and was the study approved by an Ethics Committee? Explain how the data were managed to protect confidentiality.
Please add a description of the measures taken to ensure scientific rigor. Describe the procedures implemented to ensure data quality.
Have you considered conducting additional analyses to explore possible differences between subgroups (e.g., by age, gender, level of experience with hearing aids)?

Results:The results are well presented and take into account the research questions. They use graphs and tables that facilitate reading.

Discussion:
The Discussion section is quite comprehensive and adequately addresses the results.
Ensure that all references are in the correct format (Vancouver). For example, review the reference to Fuentes-Lopez (2019) in line 523.

Include how the retrospective nature of the data and the lack of detailed demographic information could have influenced the results.

Conclusion:
The conclusion adequately summarizes the main findings. However, I recommend more explicitly highlighting how the findings can influence clinical practice and health policies. Also, reflect more deeply on the possible future impact of the findings on hearing care and how they could influence the adoption of hybrid care models.

References:
The references are fine, but I recommend reviewing them as there are some formatting errors.

I hope this helps.

Author Response

For research article

Response to Reviewer 2 Comments

1. Summary

Thank you very much for taking the time to review this manuscript. Please find our detailed responses below, as well as our corresponding revisions in the re-submitted files. We have consistently used a green font in the rebuttal letter and in the manuscript to mark our revisions and responses to the reviewer’s comments.

2. Questions for General Evaluation

Reviewer’s Evaluation

Response and Revisions

See our reply to the point-by-point responses.

3. Point-by-point response to Comments and Suggestions for Authors

Dear Authors,

I appreciate the opportunity to review your manuscript. The study addresses a relevant and highly interesting topic in the field of hearing care, especially in the context of hybrid care models. Below, I present a series of suggestions and recommendations that I believe could improve the clarity, depth, and relevance of the manuscript.

Comment 1 (Abstract):
I suggest moving the study's objective to the Introduction section of the Abstract to make the study's purpose clear from the beginning. Additionally, include a brief mention of the practical implications of the findings in the conclusion of the Abstract.

Response 1: We have moved the study’s objectives to the Introduction-section of the Abstract and have included a brief mention of the practical implications of the findings in the Conclusion of the Abstract, as per the reviewer’s suggestion.

The abstract now reads as follows: Background/Objectives: Sensorineural hearing loss (HL) is a highly prevalent chronic health condition. It can be managed through hearing care, including the use of hearing aids (HAs). Still, a majority of individuals with HL remain undiagnosed or untreated. Virtual care delivery may support uptake and adherence to interventions. In blended care, individuals can choose interchangeably between in-person and virtual services. This study aimed to investigate how real-world individuals accessed blended hearing care (through in-person, virtual, or hybrid services), the amount of support they re-ceived, and their satisfaction with services and products. Methods: An exploratory, retrospective analysis was performed on longitudinal observational data, collected through Australia’s longest running blended hearing care model. A total of 25,058 appointment records were available, matched to HA purchase records and clinical notes where possible, as well as 916 satisfaction ratings. Results: The majority of individuals attended in-person appointments (75%); 25% virtual or hybrid appointments. The number of appointments attended depended on how HAs were purchased (in-person, virtually, or hybrid), but all modalities were complemented by ample unscheduled email and telephone support. Of those who purchased HAs repeatedly, 49% changed their preferred sales channel (in-person versus virtual) over time. Satisfaction ratings were highest for virtual services. Conclusions: This first report of real-world, longitudinal evidence on blended hearing care showed strong attendance of in-person appointments, while hybrid services—including informal, unscheduled support—may have responded to individuals’ changing needs and preferences over time. The findings offer practice-based evidence for blended care models and recommendations for further research.”

Comment 2 (Introduction):
I recommend expanding the justification of the study, explaining why it is important to investigate hybrid service modalities in the context of hearing care. Please include an explanation of the practical implications of the findings. For example, how the results can influence the adoption of hybrid care models in clinical practice.
The objective is not sufficiently explicit. I recommend including it at the end of the Introduction.

Response 2: We agree with the reviewer. We added additional justifications regarding the practical implications to the introduction (p. 2, lines 82-84): “A better understanding of real-world experiences with blended care models may guide recommendations and further research aimed at increasing uptake of and satisfaction with hybrid services, ultimately supporting HA adoption overall.”

The objective has also been more explicitly stated and addressed in the Abstract and Introduction as suggested:

·       Abstract (p1, lines 26-29): “This study aimed to investigate how real-world individuals accessed blended hearing care (through in-person, virtual, or hybrid services), the amount of support they re-ceived, and their satisfaction with services and products.”

·       Introduction (p. 3, lines 116-123): During the thirteen years of clinical operation of this blended hearing care model, the longest running in Australia to date, data was collected routinely as part of ongoing service delivery. This study describes an exploratory, retrospective analysis of the observational data collected (see Methods-section for full details). Thereby, we aimed to investigate how real-world individuals accessed blended hearing care (through in-person, virtual, or hybrid services), the amount of support they received (appointments attended, as well as informal support received via email or telephone), and how individuals rated their satisfaction with blended hearing care.”

Also, we have moved the final paragraph of the Introduction to the Limitations- and Recommendations-section, following the Discussion, per the request of reviewer 3. The aim of the study, including the description of detailed research questions, is now more prominently placed at the end of the Introduction (see p. 3, lines 119-143).

Comment 3 (Introduction):

I also suggest moving any information about data sources and methodology to the Methodology section. I believe the Introduction should focus on the context, relevance of the problem, and justification of the study. All information regarding data sources should be moved to the Methodology section.

Response 3: We agree with the reviewer. We have deleted the sentence describing the data sources in the Introduction, as methodological aspects of the study are described in detail in the Methods-section.

Comment 4 (Introduction):
I suggest highlighting more clearly that the study focuses on a hybrid hearing care model, mentioning both in-person and virtual services. This should also be reflected in the objectives.

Response 4: We have rephrased the objectives of the study to include both in-person and virtual services, both in the abstract and introduction of the manuscript:

·       Abstract (p1, lines 26-29): “This study aimed to investigate how real-world individuals accessed blended hearing care (through in-person, virtual, or hybrid services), the amount of support they re-ceived, and their satisfaction with services and products.”

·       Introduction (p. 3, lines 119-123): “Thereby, we aimed to investigate how real-world individuals accessed blended hearing care (through in-person, virtual, or hybrid services), the amount of support they received (appointments attended, as well as informal support received via email or tele-phone), and how individuals rated their satisfaction with blended hearing care.”

Comment 5 (Methodology):
I suggest detailing more clearly the specific inclusion and exclusion criteria for greater transparency.

Response 5: Thank you for the suggestion. We have thoroughly reworked the Materials and Methods-section, including paragraphs describing the data sources, data cleaning, and data analysis (see p. 4-6). The section on data sources now explicitly describes which data were in- and excluded during the data export phase (see p. 4, lines 158-166): “All personal identifiers were removed from the data prior to export. Data were considered for inclusion in the data export tables based on their availability: Migration to different database systems for the Customer Management Software and HA sales database resulted in data loss, which meant that data was only available for certain periods of time depending on the data source (see Figure 1 for an overview). Data belonging to non-Australian residents were excluded (as individuals based outside of Australia did not have access to in-person services), as well as sales records that be-longed to products other than HAs (such as HA accessories).”

Comment 6 (Methodology):
More details should be provided about the statistical analysis procedures used, including any specific software.

Response 6: We have further described these details in the sections “2.1. Data Sources”, “2.2 Data Cleaning” and “2.3 Data Analysis”:

·       2.1 Data Sources (p. 4, lines 167-169): “The exported data tables were imported into R statistical software [35] and Minitab Version 17 for further data cleaning and data analysis.”

·       2.3 Data Analysis (p. 5, lines 226-237): “Data were analysed descriptively for all RQs. For RQ2a and RQ2b, hypothesis testing was employed. Based on visual inspection of the data and assumption testing, non-parametric alternatives were employed where appropriate. In RQ3, we explored satisfaction with HAs and hearing care services through ratings given on a scale from one to five. These ratings were used to calculate a Net Pro-motor ScoreTM (NPS), a metric first introduced by Reichheld [36, 37] as a means for businesses to gauge consumer loyalty [36, 37]. Individuals who provided ratings with a score of one to three were categorised as “Detractors”, four as “Passives”, and five as “Promotors”. NPS outcomes were then derived by deducting the percentage of de-tractors from the percentage of promotors [36, 37]. Confidence intervals for the NPS outcomes were calculated based on the adjusted Wald method, variation AW(3,T), as described by Rocks in 2016 [38].”

Comment 7 (Methodology):
I believe the ethical criteria of the research should be made explicit in this section. Was informed consent obtained from the participants, and was the study approved by an Ethics Committee? Explain how the data were managed to protect confidentiality. Please add a description of the measures taken to ensure scientific rigor. Describe the procedures implemented to ensure data quality.

Response 7: Thank you for the suggestion. We have thoroughly reworked the Materials and Methods-section (see p.4-6, lines 146-237), ensuring specific reference to the removal of personal identifiers (see p. 4, lines 158-159), and the data management for security (see p. 4, lines 166-167).

We have also thoroughly updated the Institutional Review Board Statement (see p. 15, lines 613-619) and Informed Consent Statement (see p. 15, lines 620-626) accordingly.

Comment 8 (Methodology):

Have you considered conducting additional analyses to explore possible differences between subgroups (e.g., by age, gender, level of experience with hearing aids)?

Response 8: We would have loved to look into this, but demographic data or data related to experience with hearing aids was not available to us as part of this dataset. The absence of demographic data in particular was related to decisions made to protect from re-identification risks, as per local ethical guidance on human research (Australian National Statement on Ethical Conduct of Human Research clause 5.1.10-5.1.18).

It is indeed a limitation and we have further addressed this in the Limitations-section of the manuscript (p. 14, lines 552-561): “It is important to highlight that the data presented here were not collected for research purposes. The data were collected routinely as part of ongoing hearing care service delivery and analysed retrospectively, which may have impacted the quality of the data and generalizability of the findings [71, 72]. Migration to new database systems over the years, for instance, led to a loss of data. Also, the dataset contained little demographic data, a limited number of therapeutic outcome measures only, and no in-formation on how often HAs were returned after purchase—all of which would be important to better understand outcomes with hybrid hearing care services. However, to the best of our knowledge, this is the first report of a large, real-world, longitudinal dataset detailing how individuals accessed and experienced blended hearing care.”

Comment 9 (Results):

The results are well presented and take into account the research questions. They use graphs and tables that facilitate reading.

Response 9: Thank you.

Comment 10 (Discussion):

The Discussion section is quite comprehensive and adequately addresses the results.
Ensure that all references are in the correct format (Vancouver). For example, review the reference to Fuentes-Lopez (2019) in line 523.

Response 10: Thank you – we have reviewed the citations and corrected the in-text citations throughout.

Comment 11 (Discussion):

Include how the retrospective nature of the data and the lack of detailed demographic information could have influenced the results.

Response 11: We have addressed this further in the Limitations-section of the manuscript (p. 14, lines 552-561): “It is important to highlight that the data presented here were not collected for research purposes. The data were collected routinely as part of ongoing hearing care ser-vice delivery and analysed retrospectively, which may have impacted the quality of the data and generalizability of the findings [71, 72]. Migration to new database systems over the years, for instance, led to a loss of data. Also, the dataset contained little demographic data, a limited number of therapeutic outcome measures only, and no in-formation on how often HAs were returned after purchase—all of which would be important to better understand outcomes with hybrid hearing care services. However, to the best of our knowledge, this is the first report of a large, real-world, longitudinal dataset detailing how individuals accessed and experienced blended hearing care.”

Comment 12 (Conclusion):
The conclusion adequately summarizes the main findings. However, I recommend more explicitly highlighting how the findings can influence clinical practice and health policies. Also, reflect more deeply on the possible future impact of the findings on hearing care and how they could influence the adoption of hybrid care models.

Response 12: We appreciate the reviewer’s request. Given the exploratory, retrospective nature of the analysis, lack of demographic information, and the fact that data were not collected for research purposes, but were collected routinely as part of an everyday clinical and business context, we want to be careful as well. We agree that the results offer practice-based evidence on experiences with blended care models. At the same time, we believe that further research is needed to influence clinical practice and health policies. Therefore, we have listed recommendations for future research (see p. 14, lines 570-582). Also, we have rephrased the final part of the conclusion (see p. 15, lines 597-601), which now states: “The findings presented offer practice-based evidence on blended care models and recommendations for further research, which may guide service delivery model design, clinical practices, and health policies. A clear recommendation to utilise and enable practice-based evidence in emerging clinical practices to inform research as it does clinical practice is of key importance.”

Comment 13 (References):
The references are fine, but I recommend reviewing them as there are some formatting errors. I hope this helps.

Response 13: Thank you – we have reviewed the citations and made the necessary corrections.

4. Response to Comments on the Quality of English Language

Point 1: The English is fine and does not require any improvement.

 Response: Thank you for the valuable comments and feedback. The manuscript has strengthened a lot based on the feedback, and we hope to have addressed the reviewer’s concerns.

Reviewer 3 Report

Comments and Suggestions for Authors

Thank you for the opportunity to review this manuscript. Please find my suggestions and comments.

Time interval included in the study

Lines 27-29:This study presents an exploratory, retrospective analysis of eight years of longitudinal data, collected through Australia’s longest running blended hearing care model.”

Lines 160-162. “The appointment records were logged between May 2011 and August 2019 and contained unique consumer codes, time stamps, scheduled appointment durations, and appointment codes.”

Lines 172-177: “While sales records were logged between July 2008 and September 2019, no sales information was available for the years prior to 2016 due to a change in database systems resulting in a loss of data during the migration. Available sales records, logged between March 2016 and September 2019, contained unique consumer codes, time stamps, and sales channel information, i.e., whether a HA was purchased in-person or virtually.”

Lines 179-181: “Between 2011 and 2021, clinicians kept track of all unscheduled support provided to individuals, i.e., support beyond appointments booked by individuals themselves.”

Lines 381-382: “In this study, we explored real-world clinical, behavioural data collected over eight  years of blended hearing care in Australia.”

It is confusing. Please, think about this and rewrite the information.  

Abstract    

Lines 31-33: Methods: “Our aims were to investigate service modality choices (in-person versus virtual), amount of support received, and satisfaction with blended hearing care” Please, the authors could include this information in “Background/Objectives”.

Introduction

-Lines 99-109: What about audiometry? Is audiometry performed to confirm the auditory thresholds of any previous exam? If so, how to perform audiometry without an acoustically isolated site in the remote context? Please, explain in the text.

-Lines 117-119: “Data were extracted retrospectively from three data sources:  (1) appointment booking software; (2) HA purchase records; and (3) clinical notes (see  Methods-section for full details).”  Please, this information is part of methodology information. It was included in “Methods” section. The authors could delete this information from “Introduction” section.

-Lines 140-147: “It is important to highlight that the data presented here were not collected for research purposes; they were routinely collected as part of ongoing service delivery. Yet, in the current era of integrating telehealth into standard healthcare practices, practice-based evidence can offer valuable insights. It has potential to identify opportunities for improvement, guide recommendations, and highlight areas for further research based on a real-world context [10, 32]. To the best of our knowledge, this is the first report of a large, real-world, longitudinal dataset detailing how individuals sought hearing care services and purchased HAs when offered fully hybrid services.”  This information could be moved to “Discussion” section.

Methods

-For a better understanding, the authors could include a figure with a summary of the steps of the methodology. Please, think about this.

-Lines 150-152: “Real-world, de-identified data were collected as part of Australia’s longest running  blended hearing care model and extracted retrospectively from three data sources: (1) appointment booking software; (2) HA purchase records; and (3) clinical notes.” I did not understand this information. Where was the data (appointment booking software , clinical notes, HA purchase records) from? From an Institute?  From a public healthcare unit? Please, explain this in the text.

-Lines 152-153:  “Prior to extraction, all data from non-Australian residents were excluded” Why? Please explain this in the text.  Remember, this is a limitation of the study that could be described in the text.

-Lines 153-157: “Due to the retrospective nature of the data analyses, lower risk of the research, removal of all personal information prior to receipt of the data by the researchers, and agreement by the researchers to take the required steps against data re-identification, no local ethics review was required for this study in Australia.” Who provided data to the authors? Could one provide data to outside researchers without the study being approved by an ethics committee? Please, think about this.

-Lines 159-160:After initial data cleaning, data of 6,864 individuals—a total of 25,058 appointment records—were available for further analysis”. Please, this information is part of results.  

-Lines 207-209:The final available dataset contained 916 service ratings and 905 product ratings: 683 records of individuals who had purchased HAs in-person and 289 records of individuals who had purchased HAs virtually.” Please, this information is part of results.  

Discussion

Line 392: “Three quarters of the 6,677 (75%; n=5,024) individuals in this sample…”  However, we can read in line 233: “results showed that 74% (n=5,024) of….”     Please, correct the information.

Author Response

For research article

Response to Reviewer 3 Comments

1. Summary

Thank you very much for taking the time to review this manuscript. Please find our detailed responses below, as well as our corresponding revisions in the re-submitted files. We have consistently used a green font in the rebuttal letter and in the manuscript to mark our revisions and responses to the reviewer’s comments.

2. Questions for General Evaluation

Reviewer’s Evaluation

Response and Revisions

See our reply to the point-by-point responses.

3. Point-by-point response to Comments and Suggestions for Authors

Thank you for the opportunity to review this manuscript. Please find my suggestions and comments.

Comment 1: Time interval included in the study

·       Lines 27-29: “This study presents an exploratory, retrospective analysis of eight years of longitudinal data, collected through Australia’s longest running blended hearing care model.”

·       Lines 160-162. “The appointment records were logged between May 2011 and August 2019 and contained unique consumer codes, time stamps, scheduled appointment durations, and appointment codes.”

·       Lines 172-177: “While sales records were logged between July 2008 and September 2019no sales information was available for the years prior to 2016 due to a change in database systems resulting in a loss of data during the migration. Available sales records, logged between March 2016 and September 2019, contained unique consumer codes, time stamps, and sales channel information, i.e., whether a HA was purchased in-person or virtually.”

·       Lines 179-181: “Between 2011 and 2021, clinicians kept track of all unscheduled support provided to individuals, i.e., support beyond appointments booked by individuals themselves.”

·       Lines 381-382: “In this study, we explored real-world clinical, behavioural data collected over eight  years of blended hearing care in Australia.”

 It is confusing. Please, think about this and rewrite the information.  

Response 1:

Thank you for pointing this out. We have thoroughly reworked the Materials and Methods-section. This section now includes Figure 1, which provides a schematic overview of what data were collected during which time frame. The Materials and Methods-section now also explicitly states which data were in- and excluded prior to the data export and its implications for the timeline (see p. 4, lines 158-166): “All personal identifiers were removed from the data prior to export. Data were considered for inclusion in the data export tables based on their availability: Migration to different database systems for the Customer Management Software and HA sales database resulted in data loss, which meant that data was only available for certain periods of time depending on the data source (see Figure 1 for an overview). Data belonging to non-Australian residents were excluded (as individuals based outside of Australia did not have access to in-person services), as well as incomplete sales records and sales records that belonged to products other than HAs (such as HA accessories).”

Figure 1. Schematic overview of data sources and timeframes during which the data were collected (left panel). Light blue boxes represent data that were unavailable for retrospective analysis due to changes in database management systems. Dark blue boxes represent data that were available for retrospective analysis. The dark green box represents data that were available, but that could not be exported automatically; instead, a manual sampling of the data was performed. The right panel provides an overview of what data were (combined and) used to address different research questions.

Comment 2: Abstract    

Lines 31-33: Methods: “Our aims were to investigate service modality choices (in-person versus virtual), amount of support received, and satisfaction with blended hearing care” Please, the authors could include this information in “Background/Objectives”.

Response 2: We agree with the reviewer. We have moved the study objective to the Introduction-section of the Abstract, which now reads as follows (see p. 1): Background/Objectives: Sensorineural hearing loss (HL) is a highly prevalent chronic health condition. It can be managed through hearing care, including the use of hearing aids (HAs). Still, a majority of individuals with HL remain undiagnosed or untreated. Virtual care delivery may support uptake and adherence to interventions. In blended care, individuals can choose interchangeably between in-person and virtual services. This study aimed to investigate how real-world individuals accessed blended hearing care (through in-person, virtual, or hybrid services), the amount of support they received, and their satisfaction with services and products. (…)”

Comment 3: Introduction

Lines 99-109: What about audiometry? Is audiometry performed to confirm the auditory thresholds of any previous exam? If so, how to perform audiometry without an acoustically isolated site in the remote context? Please, explain in the text.

Response 3: This is a very good point. For both in-person and virtual intake appointments, the blended hearing care model described relied on a triage process of reviewing clinical audiometric reports, further supported by a triage process addressing indicators of complexity, indicators of recent changes, and recommendations for further clinic-based investigation where appropriate. Discussion of virtual hearing care pathways and their clinical processes are well represented in the teleaudiology literature and beyond the scope of this paper. Exploration of the clinical process of this study however has already been detailed in Beckett et al. (2016) and Saunders et al. (2019), now added to the manuscript  with an added a sentence to clarify the process (see p. 3, line 102-103): Informational Guidance: Individuals received advice based on clinical hearing tests (for a detailed overview of the test, triage and referral procedures see [28, 29]).”

Beckett, R., B. Blamey, and S. Saunders, Optimizing Hearing Aid Utilisation using Telemedicine Tools, in Encyclopedia of E-Health and Telemedicine, M. Cruz-Cunha, et al., Editors. 2016, IGI Global: Hershey, PA, USA. p. 72-85. DOI: 10.4018/978-1-4666-9978-6.ch007

Saunders, E., S. Brice, and R. Alimoradian, Goldstein and Stephens revisited and extended to a telehealth model of hearing aid optimization, in Tele-audiology and the optimization of hearing healthcare delivery, Saunders, E. [Editor], 2019, IGI Global. p. 33-62. DOI: 10.4018/978-1-5225-8191-8.ch003

Comment 4: Introduction

Lines 117-119: “Data were extracted retrospectively from three data sources:  (1) appointment booking software; (2) HA purchase records; and (3) clinical notes (see  Methods-section for full details).”  Please, this information is part of methodology information. It was included in “Methods” section. The authors could delete this information from “Introduction” section.

Response 4: We agree with the reviewer. We have deleted the sentence describing the data sources in the Introduction, as methodological aspects of the study are described in detail in the Methods-section.

Comment 5: Introduction

Lines 140-147: “It is important to highlight that the data presented here were not collected for research purposes; they were routinely collected as part of ongoing service delivery. Yet, in the current era of integrating telehealth into standard healthcare practices, practice-based evidence can offer valuable insights. It has potential to identify opportunities for improvement, guide recommendations, and highlight areas for further research based on a real-world context [10, 32]. To the best of our knowledge, this is the first report of a large, real-world, longitudinal dataset detailing how individuals sought hearing care services and purchased HAs when offered fully hybrid services.”  This information could be moved to “Discussion” section.

Response 5: We agree with the reviewer. We have embedded this section into the Limitations- and Recommendations-section of the Discussion (p. 14, lines 552-569), which now reads as follows: “It is important to highlight that the data presented here were not collected for research purposes. The data were collected routinely as part of ongoing hearing care ser-vice delivery and analysed retrospectively, which may have impacted the quality of the data and generalizability of the findings [71, 72]. Migration to new database systems over the years, for instance, led to a loss of data. Also, the dataset contained little demographic data, a limited number of therapeutic outcome measures only, and no in-formation on how often HAs were returned after purchase—all of which would be important to better understand outcomes with hybrid hearing care services. However, to the best of our knowledge, this is the first report of a large, real-world, longitudinal dataset detailing how individuals accessed and experienced blended hearing care. The results offer practice-based evidence and may guide further research aimed at increasing uptake of and satisfaction with hybrid services, and offering more accessible and effective hearing rehabilitation services. Real-world evidence can indeed foster continuous improvements in health care provision and personalised care [73-75]. Such improvements could be aimed at adapting existing service delivery models, or designing new ones, for contemporary health consumers and the improvement of health outcomes for all. Thereby, adaptions or designs would ensure those who need help can be supported effectively, when they need it, while continuing to explore pathways to serve a larger array of individuals.”

Comment 6: Methods

For a better understanding, the authors could include a figure with a summary of the steps of the methodology. Please, think about this.

Response 6: Thank you for the suggestion. We have now introduced Figure 1, which provides a schematic overview of the data sources and timeframes during which the data were collected. Also, we have thoroughly reworked the Materials and Methods-section, starting with paragraph describing the data sources, data cleaning, and data analysis to clarify the methodology further (see p.4-6, lines 146-237).

Figure 1. Schematic overview of data sources and timeframes during which the data were collected (left panel). Light blue boxes represent data that were unavailable for retrospective analysis due to changes in database management systems. Dark blue boxes represent data that were available for retrospective analysis. The dark green box represents data that were available, but that could not be exported automatically; instead, a manual sampling of the data was performed. The right panel provides an overview of what data were (combined and) used to address different research questions.

Comment 7: Methods

Lines 150-152: “Real-world, de-identified data were collected as part of Australia’s longest running  blended hearing care model and extracted retrospectively from three data sources: (1) appointment booking software; (2) HA purchase records; and (3) clinical notes.” I did not understand this information. Where was the data (appointment booking software , clinical notes, HA purchase records) from? From an Institute?  From a public healthcare unit? Please, explain this in the text.

Response 7: Thank you for pointing out that this was unclear. All data were collected as part of ongoing service delivery, within a clinical and business context. All data were owned by the companies behind the blended hearing care model and we were granted access to the data by these companies (Blamey Saunders hears, later Sonova AG), as some of the authors of this manuscript work for the company.

We have further described this in the Methods-section (see p. 4, lines 155-156): “We were granted access to the data by the companies behind the blended hearing care model (Blamey Saunders hears, later Sonova AG).”

Comment 8: Methods

Lines 152-153: “Prior to extraction, all data from non-Australian residents were excluded” Why? Please explain this in the text.  Remember, this is a limitation of the study that could be described in the text.

Response 8: Great question. The reason for excluding non-Australian residents was twofold. On the one hand, we followed Australian ethical guidelines to conduct the study. On the other hand, non-Australian residents would not have been able to access in-person services, so the full blended hearing care model was not available to them. We have further clarified this as part of the exclusion criteria on p. 4, lines 163-166: “Data belonging to non-Australian residents were excluded (as individuals based outside of Australia did not have access to in-person services), as well as incomplete sales records and sales records that belonged to products other than HAs (such as HA accessories.

While we agree that it is a limitation, it concerned a minority of individuals (less than 5).

Comment 9: Methods

Lines 153-157: “Due to the retrospective nature of the data analyses, lower risk of the research, removal of all personal information prior to receipt of the data by the researchers, and agreement by the researchers to take the required steps against data re-identification, no local ethics review was required for this study in Australia.” Who provided data to the authors? Could one provide data to outside researchers without the study being approved by an ethics committee? Please, think about this.

Response 9: The data were provided to the authors by the companies behind the blended hearing care model, as now clarified in the text. The principle investigator was employed by the company at the time of data extraction and followed the requirements and guidance from the Australian National Statement of Ethical Conduct for Human Research. The data may be shared with outside researchers upon approval by the company.

After reworking the Materials and Methods-section, clarity is now provided regarding the access, ensuring de-identification and data protection on p. 4, lines 155-167. We have also thoroughly updated the Institutional Review Board Statement (see p. 15, lines 613-619) and Informed Consent Statement (see p. 15, lines 620-626) accordingly.

Comment 10: Methods

·       Lines 159-160: “After initial data cleaning, data of 6,864 individuals—a total of 25,058 appointment records—were available for further analysis”. Please, this information is part of results.  

·       Lines 207-209: “The final available dataset contained 916 service ratings and 905 product ratings: 683 records of individuals who had purchased HAs in-person and 289 records of individuals who had purchased HAs virtually.” Please, this information is part of results.  

Response 10: We agree with the reviewer and have removed these sentences. They were also included in the Results-section, so were redundant.

Comment 11: Discussion

Line 392: “Three quarters of the 6,677 (75%; n=5,024) individuals in this sample…”  However, we can read in line 233: “results showed that 74% (n=5,024) of….”     Please, correct the information.

Response 11: Both are correct. We started with an initial dataset of 6,864 individuals. After we had removed appointments that were cancelled or where individuals had not shown up, the dataset contained data belonging to 6,766 individuals; 74% of those (n=5,024) attended exclusively in-person appointments. In a next step of data cleaning, we also removed appointment records without appointment modality information. The final dataset belonged to 6,677 individuals, 75% of whom (n=5,024) attended in-person appointments. The exact number of individuals attending in-person appointments remained the same (n=5,024), but the proportion of individuals increased from 74% to 75% because the dataset became slightly smaller. We wanted to be fully transparent about the data cleaning steps. For the conclusion (in line 586), we referred back to the smallest dataset, following complete data cleaning.

4. Response to Comments on the Quality of English Language

Point 1: The English is fine and does not require any improvement.

Response: Thank you for the valuable comments and feedback. The manuscript has strengthened a lot based on the feedback, and we hope to have addressed the reviewer’s concerns.

Round 2

Reviewer 2 Report

Comments and Suggestions for Authors

The authors have made the suggested changes and adequately addressed the comments.

I believe the manuscript has significantly improved.

Best regards,

Reviewer 3 Report

Comments and Suggestions for Authors

Congratulations! The current manuscript is well written.